# Ammonium Ion Enhanced V_2_O_5_-WO_3_/TiO_2_ Catalysts for Selective Catalytic Reduction with Ammonia

**DOI:** 10.3390/nano11102677

**Published:** 2021-10-12

**Authors:** Min Seong Lee, Sun-I Kim, Bora Jeong, Jin-Woo Park, Taehyo Kim, Jung Woo Lee, Gibum Kwon, Duck Hyun Lee

**Affiliations:** 1Green Materials & Processes R&D Group, Korea Institute of Industrial Technology, Ulsan 44413, Korea; qneh5@kitech.re.kr (M.S.L.); sunikim@kitech.re.kr (S.-I.K.); bora1106@kitech.re.kr (B.J.); thkim0215@kitech.re.kr (T.K.); 2Department of Materials Science & Engineering, Pusan National University, Busan 46241, Korea; parkjw@nanoin.com (J.-W.P.); jungwoolee@pusan.ac.kr (J.W.L.); 3NANO. Co., Ltd., Sangju 37257, Korea; 4Department of Mechanical Engineering, University of Kansas, Lawrence, KS 66045, USA

**Keywords:** selective catalytic reduction, V-based catalyst, NO_X_ removal efficiency, silane, ammonium

## Abstract

Selective catalytic reduction (SCR) is the most efficient NO_X_ removal technology, and the vanadium-based catalyst is mainly used in SCR technology. The vanadium-based catalyst showed higher NO_X_ removal performance in the high-temperature range but catalytic efficiency decreased at lower temperatures, following exposure to SO_X_ because of the generation of ammonium sulfate on the catalyst surface. To overcome these limitations, we coated an NH_4_^+^ layer on a vanadium-based catalyst. After silane coating the V_2_O_5_-WO_3_/TiO_2_ catalyst by vapor evaporation, the silanized catalyst was heat treated under NH_3_ gas. By decomposing the silane on the surface, an NH_4_^+^ layer was formed on the catalyst surface through a substitution reaction. We observed high NO_X_ removal efficiency over a wide temperature range by coating an NH_4_^+^ layer on a vanadium-based catalyst. This layer shows high proton conductivity, which leads to the reduction of vanadium oxides and tungsten oxide; additionally, the NO_X_ removal performance was improved over a wide temperature range. These findings provide a new mothed to develop SCR catalyst with high efficiency at a wide temperature range.

## 1. Introduction

Most energy is generated through the combustion of fossil fuels, which causes air pollution by emitting toxic pollutants, such as NO_X_, particulate matter, SO_X_, and CO [1,2]. Nitrogen oxides (NO, NO_2_, and N_2_O) are the most representative air pollutants because they are major causes of acid rain, photochemical smog, ozone depletion, and global warming [3,4,5]. NO_X_ is emitted by automobiles, power stations, industrial heaters, and non-road vehicles. To reduce this form of pollution, NO_X_ removal technologies must be developed. Many technologies have been developed to control NO_X_ emissions, such as selective catalytic reduction (SCR), non-selective catalytic reduction (NSCR), and selective non-catalytic reduction (SNCR). Selective catalytic reduction (SCR), the most efficient NO_X_ removal technology, selectively converts NO_X_ in fuel gas into N_2_ and H_2_O. Compared with other technologies, such as non-selective catalytic reduction (NSCR) and selective non-catalytic reduction (SNCR), SCR shows a high NO_X_ removal efficiency and no secondary pollutant emission and can be operated in the temperature range of 100–300 °C [4].

The V_2_O_5_-WO_3_/TiO_2_ catalyst has been commercially applied for SCR of NO_X_ with NH_3_ because of its good thermal stability, N_2_ selectivity, and low SO_2_ oxidation activity for conversion to SO_3_ [6]. However, this catalyst shows high NO_X_ removal performance only in the temperature range of 300–400 °C, and performance decreases at temperatures below 300 °C [7,8]. Therefore, many studies have been conducted to develop catalysts that function over a wide temperature range such as 200–420 °C. Djerad et al. increased the loading of V_2_O_5_ in WO_3_/TiO_2_ catalysts to improve NO_X_ removal efficiency at lower temperatures [9,10]. A higher V_2_O_5_ content leads to increased catalytic activity but decreased N_2_ selectivity and a narrow activation temperature range. Other researchers modified catalysts using transition metal oxides (Co, Mn, Ce, Cu, etc.) and rare-earth oxides (Eu, Sm, Nd, and Ho) [3,11,12,13,14,15,16,17,18,19]. For example, Wang et al. investigated the effect of adding Cu to a V_2_O_5_-WO_3_/TiO_2_ catalyst on low-temperature NH_3_-SCR performance [20]. The added Cu oxides improved the catalytic activity of V_2_O_5_ by increasing the redox property of V_2_O_5_ (Cu^2+^ + V^4+^ ⇄ V^5+^ + Cu^+^), leading to a fast SCR reaction. GuO et al. modified Nd on MnOx catalysts to improve NH_3_ SCR performance [21]. The catalytic activity and N_2_ selectivity were enhanced by doping MnOx with an Nd catalyst. Although catalytic activity was improved at low temperatures (>300 °C), the catalytic performance decreased at temperatures above 400 °C. Therefore, covering a wide window temperature by modifying only the catalyst remains challenging. Another method for improving NO_X_ removal technology at wide temperature ranges (200–420 °C) involves applying ammonium nitrate (NH_4_NO_3_) on the catalysts [22,23,24]. Chae et al. enhanced the catalytic performance of SCR of NO_X_ at temperatures below 300 °C by injecting liquid ammonium nitrate on a V_2_O_5_-Sb_2_O_3_/TiO_2_ catalyst. The injected ammonium nitrate led to a fast SCR and prevented the formation of ammonia bisulfate on the catalyst surface and catalytic deactivation by SO_2_ at temperatures below 300 °C.

In this study, we enhanced the NO_X_ removal efficiency and N_2_ selectivity over a wide temperature range of 200–420 °C by improving the catalytic activity at temperatures below 300 °C. The catalyst surface was coated with a silane layer, and the silane molecules evaporated with heat treatment at 500 °C in NH_3_ environment. The resulting catalyst surface was substituted with NH_4_^+^ layers present, and the NH_4_^+^ layers formed on the surface of the V_2_O_5_-WO_3_/TiO_2_ catalyst increased NO_X_ removal efficiency at low temperatures (200–250 °C).

## 2. Materials and Methods

### 2.1. Catalyst Preparation

The NH_4_^+^-coated V_2_O_5_-WO_3_/TiO_2_ catalyst was prepared in three steps: first, V_2_O_5_-WO_3_/TiO_2_ was synthesized by the impregnation method, and then, the catalyst underwent vapor phase silane treatment at 180 °C for 1 h. Finally, the silanized V_2_O_5_-WO_3_/TiO_2_ catalysts were heat treated under NH_3_ gas with N_2_ gas.

To synthesize 2 wt.% V_2_O_5_ −10 wt.% WO_3_/TiO_2_, 8.800 g of TiO_2_ powder (8.800 g, NANO Co., Ltd., Seoul, Korea, NT-01), 0.256 g of NH_4_VO_3_ (Sigma-Aldrich, St. Louis, MO, USA, 99.99%), 1.062 g of (NH_4_)_6_H_2_W_12_O_40_ × H_2_O (Sigma-Aldrich, St. Louis, MO, USA, 99.99%), and 0.386 g of oxalic acid were mixed with 100 mL of deionized water and stirred for 2 h. The solution in the mixture was evaporated at 85 °C in an oil bath for impregnation of V and W atoms followed by drying in an oven at 110 °C for 12 h. The dried powder was placed in a furnace and calcinated at 500 °C under atmospheric pressure.

Vapor phase silanization was conducted. First, the catalyst was cleaned by ozone treatment to remove any impurities and to increase hydroxyl species on the surface. The catalyst was placed in a sealed Teflon jar with 1.0 mL of Trichloro (1H, 1H, 2H, 2H-perfluorooctyl) silane (Sigma-Aldrich, 97%). The jar was subjected to 180 °C to vary the time. To coat the NH_4_^+^ on the catalyst, the silanized V_2_O_5_-WO_3_/TiO_2_ catalyst underwent heat treatment at 500 °C for 2 h upon purging a of NH_3_ gas (300 ppm) for 2 h.

To observe the effect of the NH_4_^+^ layer on the commercial catalyst, we prepared commercial plate-type monoliths. The catalyst, consisting of V_2_O_5_, MoO_3_, and TiO_2_, was pasted on the wall of the plate, which had dimensions of 1mm × 34mm × 187 mm, and an NH_4_^+^ layer coating was performed. After silane coating by vapor deposition, the catalysts were heat treated at 500 °C under 300 ppm NH_3_ gas with N_2_ gas for 2 h.

### 2.2. Catalyst Characterization

The effect of each preparation step on the catalyst morphology was investigated by field emission scanning electron microscopy (Hitachi, Tokyo, Japan, SU8020) at an accelerating voltage of 15.0 kV. The surface properties were determined by measuring the water contact angle using the static sessile drop method with a Contact angle analyzer (Phoenix 300, SEO, Suwon, Korea). The contact angle of the prepared catalysts (thickness = about 0.2 cm) was measured within 10 s after dropping water

The crystallinity of and impurities in the catalysts were analyzed using X-ray diffraction (XRD, Rigaku, Tokyo, Japan, Ultima IV) with Cu Kα (λ = 0.15406 nm) radiation in the 2θ range of 10° to 80° at a scan rate of 1°/min. The Si and N elements in the catalysts were measured by X-ray photoelectron spectroscopy (XPS; Thermo VG Scientific, Waltham, MA, USA, K Alpha+) with Al Kα radiation; the binding energy of C1s was normalized to 284.8 eV. Fourier transform infrared spectroscopy (FT-IR, Varian Medical Systems, Palo Alto, CA, USA, 670 FTIR) was carried out over a wavelength range of 400–4000 cm^−1^.

H_2_-temperature-programmed reduction was carried out using an AutoChem II 2920 (Micromeritics Instrument Corp., Norcross, GA, USA). The samples were exposed to a current of 10% H_2_/Ar and measured in the 200–900 °C temperature range.

### 2.3. Catalytic Activity Measurement

The NO_X_ removal efficiency of the catalysts was evaluated in a fixed-bed reactor under high atmospheric pressure, with the sample placed on a stainless-steel tube and analyzed using a powder catalyst. The analysis temperature was varied from 150 °C to 420 °C, and the reactive gas was composed of 300 ppm NO_X_, 300 ppm NH_3_ (NH_3_/NO_X_ = 1.0), 300 ppm SO_2_, and 5 vol.% of O_2_, with a N_2_ balance under a total flow rate of 500 sccm. The powder catalyst (0.35 g) was tested, and the gas hourly space velocity was set at 60,000 h^−1^. Plate-type commercial catalysts were evaluated in a microreactor. The analysis temperature was varied from 200 °C to 450 °C, and the reactive gas was composed of 240 ppm NO_X_, 288 ppm NH_3_ (NH_3_/NO_X_ = 1.2), 600 ppm SO_2_, and 3 vol.% of O_2_, with a N_2_ balance under a total flow rate of 1.2 m^3^/h; the area velocity was set to 25 m/h. The reactive gas concentration was determined by FT-IR (Gasmet, Vantaa, Finland, CX-4000) and gas analysis spectrometry (DSMXT, Heidelberg, Germany). NO_X_ removal efficiency and N_2_ selectivity were calculated according to Equations (1) and (2), respectively.
(1)$${\mathrm{NO}}_{X}\;\mathrm{removal}\;\mathrm{efficiency}\;\left(\%\right)=\frac{{\mathrm{NO}}_{X\;\mathrm{inlet}}-{\mathrm{NO}}_{X\;\mathrm{outlet}}}{{\mathrm{NO}}_{X\;\mathrm{inlet}}}\times 100,$$
(2)$$N_{2}\;\mathrm{selectivity}\;\left(\%\right)=\frac{{\mathrm{NO}}_{\;\mathrm{inlet}}-{\mathrm{NO}}_{\;\mathrm{outlet}}-{\mathrm{NO}}_{2\;\mathrm{outlet}\;}-{\;N}_{2}O_{\mathrm{outlet}}}{{\mathrm{NO}}_{\;\mathrm{inlet}}-{\mathrm{NO}}_{\;\mathrm{outlet}}}\times 100,$$

## 3. Results

Figure 1 illustrates the process of producing the NH_4_^+^ layer on the V_2_O_5_-WO_3_/TiO_2_ catalysts. The surface of the synthesized V_2_O_5_-WO_3_/TiO_2_ catalysts is hydrophilic because of the presence of numerous oxygen groups. After the catalysts were silanized at 180 °C for 1 h, the surface properties became hydrophobic because of the hydrophobic agent in silane [25]. When the catalysts were heat treated at 500 °C under 300 ppm NH_3_ gas with N_2_ gas for 2h, the silane molecules underwent thermal decomposition. Subsequently, the catalyst surface acquired NH_4_+ on their active sites. As a result, the V_2_O_5_-WO_3_/TiO_2_ catalysts were covered with NH_4_^+^ layers, generating hydrophilic surface properties.

The surface properties and morphology of the prepared catalysts were investigated using a scanning electron microscopy, and contact angle measurements. Figure 2a,d show the V_2_O_5_-WO_3_/TiO_2_ catalysts without treatment. Figure 2b,c,e,f show the results obtained for the catalyst after silane treatment and the following heat treatment, respectively. To observe the surface properties, we dropped water on the surface of the prepared SCR catalyst powders; the microscope observation results are shown in Figure 2a–c. Water that dropped on the surface was immediately absorbed into the original V_2_O_5_-WO_3_/TiO_2_ catalyst. The inset image shows the results of static contact angle measurement. The observed contact angle was approximately 2°, indicating that the catalysts were hydrophilic. After silane treatment of the V_2_O_5_-WO_3_/TiO_2_ catalyst, the water did not spread on the surface but rather formed a drop on the surface with a contact angle of approximately 160°. As the hydrophobic agent in silane covered the catalyst, the surface became hydrophobic [26]. However, the surface properties became hydrophilic after heat treating the catalyst at 500 °C for 2h under NH_3_ gas. This is because the silane molecules are thermally decomposed at a high temperature (e.g., 500 °C). Subsequently, NH_3_ can react to the active sites of catalyst and form NH_4_^+^ on the surface as a monolayer. Although the surface properties were changed by each process, the catalyst morphologies (Figure 2d–f), catalyst contents, and the textural properties (Table 1) were not changed because the prepared silane and NH_4_^+^ layers are quite thin.

The crystalline structures of the prepared catalysts were determined using XRD (Figure 3a), which clearly showed anatase TiO_2_, whereas peaks for V_2_O_5_ and WO_3_ were not observed in any samples. This is because the V_2_O_5_ and WO_3_ catalysts with small nanoparticle sizes were uniformly coated on the TiO_2_ support, and the peaks of the V_2_O_5_ and WO_3_ catalysts were hard to detect using XRD [27]. Furthermore, the peak intensities and positions were similar for all catalysts, indicating that silane and heat treatments under NH_3_ gas did not affect the crystallinity of the V_2_O_5_-WO_3_/TiO_2_ catalyst.

To observe the silane layer and difference in chemical bonding in the catalyst after the heat-treatment process, XPS analysis was conducted (Figure 3b–d). Figure 3b shows the survey peaks in the XPS spectra. The XPS spectra of the Si 2p peak of the V_2_O_5_-WO_3_/TiO_2_ catalyst (black line), silanized V_2_O_5_-WO_3_/TiO_2_ catalyst (blue line), and following catalyst after heat treatment (red and green lines) are shown in Figure 3c. In the V_2_O_5_-WO_3_/TiO_2_ catalyst, no Si 2p peak was observed. After silane treatment, silicon atoms appeared at low binding energy (102 eV), indicating that silane was coated on the V_2_O_5_-WO_3_/TiO_2_ catalyst. However, the peaks at 102 eV disappeared after heat treatment of the silanized V_2_O_5_-WO_3_/TiO_2_ catalyst, indicating that the coated silane had decomposed at temperatures over 500 °C. By contrast, binding energies near 401 and 399 eV were observed for the heat-treated V_2_O_5_-WO_3_/TiO_2_ catalyst (Figure 3d). The intensity of V_2_O_5_-WO_3_/TiO_2_ catalyst and silanized V_2_O_5_-WO_3_/TiO_2_ catalyst were lower, but the intensity was larger for the heat-treated V_2_O_5_-WO_3_/TiO_2_ catalyst. The binding energies were attributed to the neutral amine’s residual on the surface and NH_4_^+^, respectively, from the flowing NH_3_ gas during the heat-treatment process [28,29]. The NH_2_ peak at around 399 eV came from the use of ammonium-based precursors of V and W, and the NH_2_ peak is observed in all samples. Table 2 and S1 show the atomic contents of the prepared catalysts obtained from XPS spectra. Heat-treated V_2_O_5_-WO_3_/TiO_2_ catalyst had increased N concentration of 0.79 at.% compared with the V_2_O_5_-WO_3_/TiO_2_ catalyst (0.40 at.%) and the Silanized V_2_O_5_-WO_3_/TiO_2_ catalyst (0.31 at.%). The results indicate that the NH_4_^+^ layer is formed on the heat-treated V_2_O_5_-WO_3_/TiO_2_ catalyst.

The NO_X_ removal efficiency and N_2_ selectivity following the formation of the NH_4_^+^ layer on the surface of the V_2_O_5_-WO_3_/TiO_2_ catalyst were measured in a fixed bed (Figure 4). Figure 4a shows the NO_X_ removal efficiency of the prepared samples, which were higher for the silanized catalyst and heated silanized catalyst than for the original V_2_O_5_-WO_3_/TiO_2_ catalyst below 300 °C. The catalytic performance of the silanized V_2_O_5_-WO_3_/TiO_2_ catalyst improved from 45% to 60%, particularly at 200 °C. Figure 4b shows the N_2_ selectivity; a trace amount of N_2_O in the prepared catalyst was produced at 300 °C. The trend in N_2_ selectivity was similar to the NO_X_ removal efficiency, and the heat-treated V_2_O_5_-WO_3_/TiO_2_ catalyst showed a higher N_2_ selectivity at 420 °C. Sulfur in the fuel deactivated the catalyst through the poisoning effect, specifically in the low-temperature range (below 300 °C). When vanadium catalysts are exposed to SO_x_, ammonium bisulfate (NH_4_HSO_4_) and ammonium sulfate ((NH_4_)_2_SO_4_) are formed on the catalyst surface, decreasing efficiency [24]. The hydrophobic properties of the coated silane prevent the formation of ammonium sulfate on the V_2_O_5_-WO_3_/TiO_2_ catalyst surface, enhancing the NO_X_ removal efficiency at low temperatures. Interestingly, the NO_X_ removal efficiency of the silanized catalyst after heat treatment was higher than that of the silanized and original catalysts. The NH_4_^+^ intermediates in the catalyst exhibit high catalytic activity and proton conductivity in NH_3_ SCR, particularly in the low-temperature range. Therefore, the NH_4_^+^ layer on the catalyst produced by silane and heat treatments leads to enhanced NO_X_ removal efficiency at low temperatures, specifically at 250 °C, because of the higher proton conductivity and reactivity [30], whereas silane is decomposed and reacts with NH_3_ at over 300 °C in the reaction gas during fixed bed evaluation, and some NH_4_^+^ is produced on the catalyst surface. Therefore, the silanized and heat-treated V_2_O_5_-WO_3_/TiO_2_ catalysts show similar catalytic performances at high temperatures (>300 °C). Note that there was no significant difference in the NO_X_ removal efficiency and N_2_ selectivity between V_2_O_5_-WO_3_/TiO_2_ and heat-treated V_2_O_5_-WO_3_/TiO_2_ without a silane coating process in the temperature range of 150–420 °C (Appendix A). This results demonstrate that silane coating is a key process to prepare ammonium ion enhanced V_2_O_5_-WO_3_/TiO_2_ catalysts.

The formed NH_4_^+^ layer and acid sites were analyzed using FT-IR spectra, as shown in Figure 5a–c. Si-CH_2_ and Si-O-Si bond peaks only appeared at 1250 and 1216 cm^−1^, respectively, in the silanized V_2_O_5_-WO_3_/TiO_2_ catalyst (Figure 5b). After heat treatment, the peaks related to Si (e.g., 1250 and 1216 cm^−1^) disappeared while various peaks related to NH_4_^+^ started to appear in the range of 1400 to 1600 cm^−1^. Furthermore, the peak near 3300 cm^−1^ corresponds to N–H stretching (Figure 5c). According to these results, the coated silane layer disappeared during heat treatment, and a new NH_4_^+^ layer was generated after the decomposition of Si atoms on the surface. To investigate the redox properties of the prepared catalysts depending on the silane treatment and formation of the NH_4_^+^ layer, we analyzed the H_2_-temperature-programmed reduction profiles (Figure 5d). V_2_O_5_-WO_3_/TiO_2_ exhibited three reduction peaks at 397.1 °C, 480.2 °C, and 800.8 °C, which were assigned to the reduction of V^5+^ to V^3+^ in the vanadium oxides, the reduction of W^6+^ to W^4+^, and the reduction of W^4+^ to W^0^ in the tungsten oxide, respectively [31,32,33]. After coating the NH_4_^+^ layer on the V_2_O_5_-WO_3_/TiO_2_ catalyst surface, the reduction peaks shifted to lower temperatures of 385.6 °C, 461.2 °C, and 778.7 °C, respectively. As the NH_4_^+^ layer on the surface has high proton conductivity and activity, the active materials reduced a larger amount of NO_X_.

To investigate the effect of the NH_4_^+^ layer under actual conditions, we applied the NH_4_^+^ layer on the commercial catalyst (Figure 6). Figure 6a shows a plate-type commercial catalyst containing V_2_O_5_, MoO_3_, TiO_2_, etc. (Appendix A), which is used in electric power stations to reduce NO_X_ gas emissions. The catalyst surface was hydrophilic, similar to the power type of the catalyst (Figure 6b). Silane treatment was conducted by the vapor evaporation method, and the surface became hydrophobic (Figure 6c). Finally, after heat treating the silanized catalysts at 500 °C for 2h under NH_3_ gas with N_2_ gas, the surface properties were hydrophilic because of decomposition of the silane layer and because of coating the NH_4_^+^ layer on the catalyst surface (Figure 6d).

The prepared plate-type commercial catalysts were evaluated in a microreactor from 200 °C to 450 °C, as shown in Figure 7. The commercial catalysts exhibited a lower NO_X_ removal efficiency at 200 °C, which gradually increased to 400 °C. The commercial catalyst specialized in high temperature range over 300 °C because it used an electric power plant. Therefore, the NO_X_ removal efficiency is not largely changed at high temperature. After forming the NH_4_^+^ layer on the catalyst, the NO_X_ removal efficiency was enhanced over a wide window range from 200 °C to 450 °C, particularly at 200 °C. Usually, the catalytic efficiency deteriorated at lower temperature due to generating ammonium sulfate on the catalyst surface. However, the formed NH_4_^+^ layer has high proton conductivity, which leads to the reduction of vanadium oxides and tungsten oxide, and improved the NO_X_ removal efficiency.

## 4. Conclusions

We investigated the effect of coating of V_2_O_5_-WO_3_/TiO_2_ catalysts with an NH_4_^+^ layer through heat treatment of silanized catalysts at 500 °C under NH_3_ gas with N_2_ gas. The silane molecules were substituted with NH_4_^+^ by decomposing the silane layer on the surface. V_2_O_5_-WO_3_/TiO_2_ catalysts with an NH_4_^+^ layer showed an increased NO_X_ removal efficiency over a wide temperature range (150–420 °C) compared with the V_2_O_5_-WO_3_/TiO_2_ catalysts. Particularly, the performance was improved at low temperatures (<300 °C). The coated NH_4_^+^ layer led to enhanced proton conductivity and promoted further reduction of vanadium oxides and tungsten oxide. The V_2_O_5_-WO_3_/TiO_2_ catalysts with an NH_4_^+^ layer exhibited improved SCR performance. These results may contribute to future studies on SCR catalysts and other catalyst systems.

## Figures and Tables

**Figure 1 nanomaterials-11-02677-f001:**
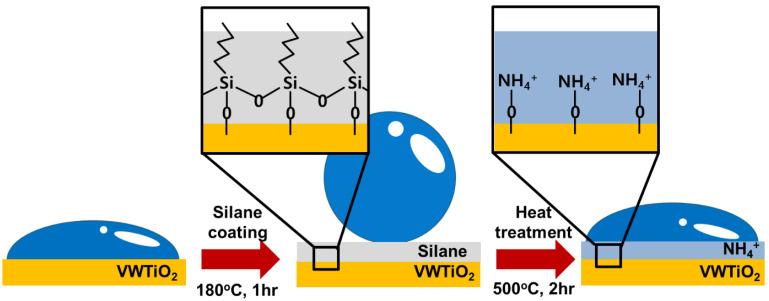
Schematic illustration of V_2_O_5_-WO_3_/TiO_2_, silanized V_2_O_5_-WO_3_/TiO_2_, and heat-treated V_2_O_5_-WO_3_/TiO_2_ with NH_4_^+^ layer coating.

**Figure 2 nanomaterials-11-02677-f002:**
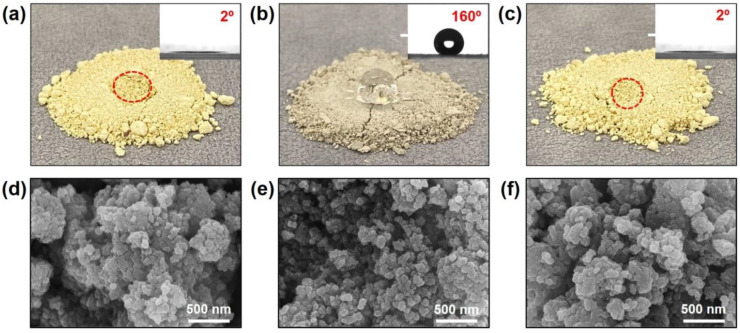
Microscope images of V_2_O_5_-WO_3_/TiO_2_ (**a**), silanized V_2_O_5_-WO_3_/TiO_2_ (**b**), and heat-treated V_2_O_5_-WO_3_/TiO_2_ (**c**) (insets are contact angle measurement). The red circle marks the location where the water is absorbed on the catalyst. SEM images of V_2_O_5_-WO_3_/TiO_2_ (**d**), silanized V_2_O_5_-WO_3_/TiO_2_ (**e**), and heat-treated V_2_O_5_-WO_3_/TiO_2_ (**f**).

**Figure 3 nanomaterials-11-02677-f003:**
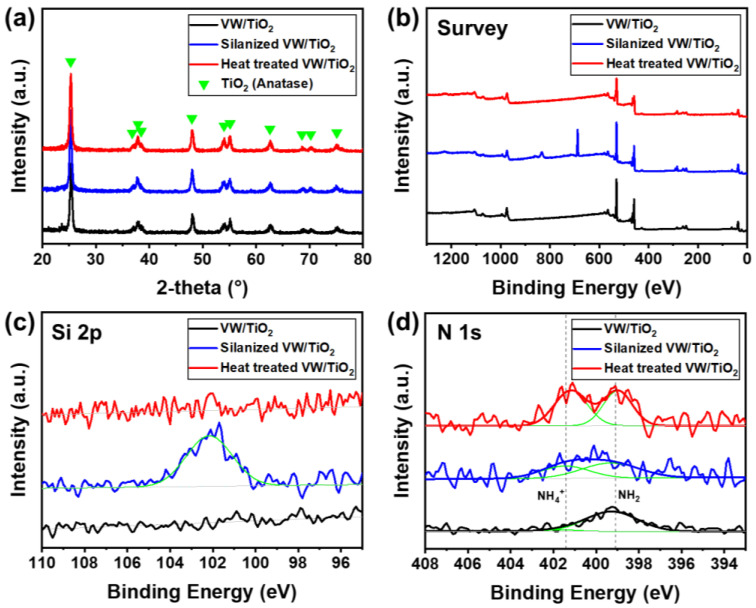
(**a**) XRD patterns and XPS spectra for (**b**) survey, (**c**) Si 2p, and (**d**) N 1s of V_2_O_5_-WO_3_/TiO_2_ (black line), silanized V_2_O_5_-WO_3_/TiO_2_ (blue line), and heat-treated V_2_O_5_-WO_3_/TiO_2_ (red line).

**Figure 4 nanomaterials-11-02677-f004:**
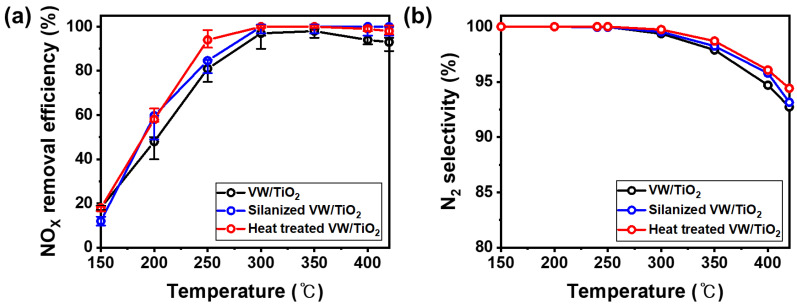
NOx removal efficiency (**a**) and N_2_ selectivity (**b**) of V_2_O_5_-WO_3_/TiO_2_ (black line), silanized V_2_O_5_-WO_3_/TiO_2_ (blue line), and heat-treated V_2_O_5_-WO_3_/TiO_2_ (red line). Reaction conditions: [NO] = 300 ppm, [NH3] = 300 ppm, [SO_2_] = 600 ppm, [O_2_] = 5 vol.%, and [gas hourly space velocity] = 60,000 h^−1^.

**Figure 5 nanomaterials-11-02677-f005:**
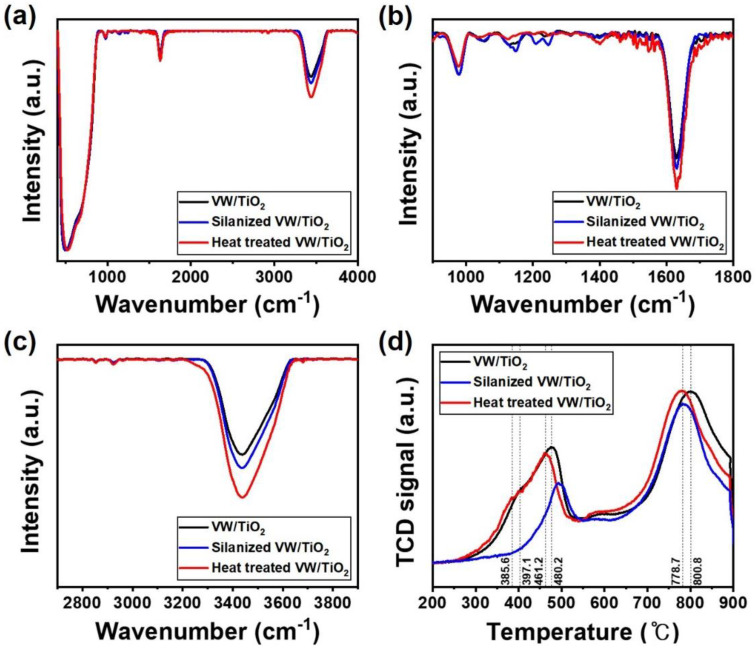
Fourier transform infrared spectra (**a**–**c**) and H_2_-temperature-programmed reduction (**d**) of V_2_O_5_-WO_3_/TiO_2_ (black line), silanized V_2_O_5_-WO_3_/TiO_2_ (blue line), and heat-treated V_2_O_5_-WO_3_/TiO_2_ (red line).

**Figure 6 nanomaterials-11-02677-f006:**
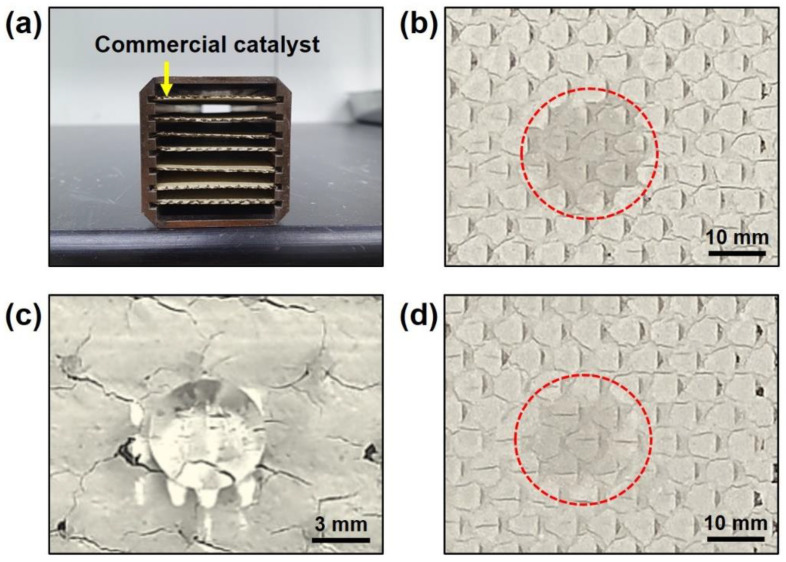
Microscope image of plate-type commercial catalyst (**a**), commercial catalyst surface (**b**), silanized commercial catalyst surface (**c**), and heat-treated commercial catalyst surface (**d**) after dropping water.

**Figure 7 nanomaterials-11-02677-f007:**
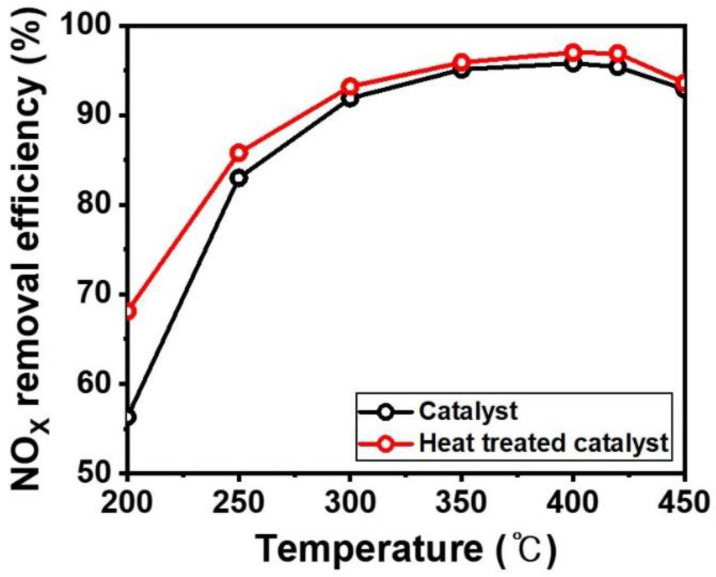
NOx removal efficiency of V_2_O_5_-WO_3_/TiO_2_ (black line) and heat-treated V_2_O_5_-WO_3_/TiO_2_ (red line). Reaction conditions: [NO] = 240 ppm [NH3] = 288 ppm, [SO_2_] = 600 ppm, [O_2_] = 3 vol.%, [area velocity] = 25 m/h.

**Table 1 nanomaterials-11-02677-t001:** X-ray fluorescence analysis and Brunauer-Emmet-Teller (BET) results of the V_2_O_5_-WO_3_/TiO_2_, silanized V_2_O_5_-WO_3_/TiO_2_, and heat-treated V_2_O_5_-WO_3_/TiO_2_ catalyst.

Sample	Chemical Composition (wt.%)				S_BET_ (m^2^/g)	Pore Volume (cm^3^/g)	Pore Size (nm)
	TiO_2_	V_2_O_5_	WO_3_	SO_3_			
VW/TiO_2_	87.53	2.03	10.04	0.40	68.86	0.28	15.94
Silanized VW/TiO_2_	87.32	2.05	10.02	0.61	65.54	0.21	12.95
Heat-treated VW/TiO_2_	87.53	1.98	9.99	0.50	66.39	0.24	14.17

**Table 2 nanomaterials-11-02677-t002:** Atomic percent of element obtained from XPS spectra of the V_2_O_5_-WO_3_/TiO_2_, silanized V_2_O_5_-WO_3_/TiO_2_, heat-treated V_2_O_5_-WO_3_/TiO_2_ catalyst.

Sample	Element (at.%)								
	N	O	Si	F	C	S	V	W	Ti
VW/TiO_2_	0.40	52.47	-	-	18.28	0.53	1.55	1.82	24.95
Silanized VW/TiO_2_	0.31	45.90	0.24	12.89	19.91	0.24	1.28	1.36	17.89
Heat-treated VW/TiO_2_	0.79	56.44	-	-	18.06	0.61	1.34	1.80	20.96

## Data Availability

The data are contained within the article.

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
