# Peer review of "Ammonium Ion Enhanced V2O5-WO3/TiO2 Catalysts for Selective Catalytic Reduction with Ammonia"

_nanomaterials, 2021, doi:10.3390/nano11102677_

Round 1

Reviewer 1 Report

The paper reports the SCR reaction using VOx-WOx-TiOx catalysts that have been treated with silane and then NH3. The authors propose the formation of a NH4+ surface layer which enhances proton conductivity and hydrophilicity leading to higher selectivity and activity. As the main hypothesis of the paper is the formation of this NH4+ layer I find the explanation and evidence somewhat lacking in addition to missing crucial control reactions. For this reasons I do not recommend publication unless major points are addressed.

Firstly – a crucial control is missing – the treatment of the fresh catalyst – before silane – under the same conditions the authors report to give high activity – ammonia at high temperature.

Second – the discussion of the NH4+ layer is confusing – the authors suggest an ionic surface species resulting from silane decomposition – but in fact it should have a counter anion. Is this a simple acid base reaction with surface acid species? Reaction with VOx or WOx compounds to from ammonium vanadate or tungstate? An surface acid-base type reaction with for example Ti-OH to form Ti-O-NH4 would result likely lead to ammonia evolution at high temperatures akin to a NH3 TPD experiment.  

Thirdly the XPS – the authors show a N+ species at 400 eV – what is this? In addition, the two N species implicated in high activity are present in the fresh sample? Is this relative small change in concentration responsible for high activity? The XPS peak for the ammonium is a little lower than expected for ammonium salts such as ammonium chloride – can any information about the nature of the species be determined – XPS of V and W species could provide information on the surface reactions occurring.

Author Response

We appreciate reviewer’s valuable comments. We have now modified the manuscript, so that the readers can understand the main points of our manuscript.

Reviewer 2 Report

  • How does the author decided 2 wt.% V2O5-10 wt.% WO3 is an appropriate loading to work with? Any reason or preliminary data to support this?
  • What is the loading of V2O5 and WO3 on the commercial plate-type monoliths?
  • Does the loading of the chosen V2O5 and WO3 loading match the commercial monoliths?
  • Ozone treatment led to the increase of hydroxyl species on the surface (line 91) and making the surface of the synthesized V2O5/WO3/TiO2 hydrophilic (line 141), any other treatment to use to remove the impurities and not introduce hydroxyl species on the surface?
  • How does the concentration of NH3 gas decided?
  • Does higher concentration of NH3 has thicker layer of NH4+?
  • How thick is the NH4+ layer on the catalyst at 300 ppm? Elemental analysis may able to provide information on the atomic percentage of % N and H% and thus calculate the thickness of NH4+
  • Does the thickness of NH4+ layer affect the catalytic activity of V2O5/WO3/TiO2?
  • More characterization analysis needed on the NH4+ layer as this is the highlight of this work
  • Format issue, the 4 in NH4+ in Figure 1 need to be subscripted.
  • Is the hydrophobic and hydrophilic testing reliable on powder samples? The result will strongly dependent on the displacement of the powder samples. How many times did the author perform the contact angle testing, and is it reproducible? Please provide more experimental details in the catalyst characterization session.
  • Does the catalyst particle size increases after deposition of silane and NH4+ layer? Figure 2(f) seems to have larger particles after heat treatment.
  • According to Figure 3 (d), N species were present even before the silane and NH4+ layer deposition, how the author explains the present of N in the as prepared V2O5/WO3/TiO2? The presence of N species was also found in silanized V2O5-WO3/TiO2 catalyst, any reason?
  • Please provide clear figure caption label to Figure 3 (c) and (d).
  • The XPS fitting needs to be improved as the peak position of N+ and NH4+ aren’t align and the FWHM are very different as well.
  • Comparison of characterization results of V2O5/WO3/TiO2 prepared in this work with commercial plate-type monoliths would be good.
  • Can author identify the thickness of NH4+ layer? Does NH4+ layer thickness affect the NOx removal efficiency?
  • 450 oC should be clearly labelled in Figure 4(a) and (b).
  • How many time does the author perform the NOx removal efficiency testing?
  • Could the author provide error bars on Figure 4? Silanized VW/TiO2 seems have similar performance to heat treated VW/TiO2, they might be within its error bar range? Please provide more solid experiments data to prove the deposition of NH4+ layer is necessary in VW/TiO2
  • The deposition of NH4+ layer doesn’t have much effect on improving the efficiency based on conclusion from Figure 4 and S1. Again, any needs to deposition NH4+ layer and how this NH4+ layer play its role to improve the efficiency and selectivity?
  • Format issue, the 4 in NH4+ in line 234 need to be subscripted.
  • Figure 5 (c) showed be stated in line 236.
  • Can H2-temperature-programmed reduction profiles provide atomic percentage of N, O and H in the VW/TiO2 system?
  • Again, the NOx removal efficiency of commercial samples with and without NH4+ layer are very similar from 300-450 o Please provide more solid experiments to prove that NH4+ layer is critical to the NOx removal efficiency.

Author Response

(The authors gave the same response as above.)

Round 2

Reviewer 1 Report

The points raised have now been addressed and i have no further comments to add. 

Author Response

We appreciate the reviewer's comment and do not have any reply to the reviewer.

Reviewer 2 Report

Few minor suggestions: 

  • Please provide units (e.g. wt.% or at.%) in X-ray fluorescence Table 1 and Table S2.
  • Please state clearly what the author mean by low temp and high temperature. Very inconsistent use of these term which lead to the confusion.  Example: 

    Line 4, 350 oC was stated as relatively low temperature, however, in the coverletter, the author state “ removal efficiency at high temperature (300 – 400 oC)”. The use of high and low temperature is not clear and need to be improved.

  • Please clearly state the purpose of the research in the introduction section, such as improvement of the catalytic performance at low temperature (e.g. >250 oC)”.

    Again, please watch out the use of “high and low”. Please go through the induction section and make it clear. Please mention the actual temperature used in the reference [9] –[24], rather than just use “high temp, low temp, lower temp “ to summarized the literatures.

    The term “wide range of temperature” and “low temperature ~350 oC”, high temperature (300-400 oC) is very misleading in the introduction and does not show the target temperature range (< 250 oC). The goal of this research at low temperature (e.g. < 250 oC) should be stated in the introduction.

  • Line 76-77, “Finally, NH4+ layers formed on the surface of the V2O5-WO3/TiO2 catalyst, which increased NOx removal efficiency over a wide temperature range.” Should be changed to “Finally, NH4+ layers formed on the surface of the V2O5-WO3/TiO2 catalyst, which increased NOx removal efficiency at low temperature (~200-250 oC).”

One major suggestion: 

  • The discussion in Question 20 with the possible formation of NH4+ layer on the silane coated samples should be included in the main text and the XPS analysis of post-tested silane coated sample should be performed and presented to convenience the readers. Otherwise, the result suggested silane coating seems more critical to improve the NOx removal efficiency over a "wide range" of temperature. Addition of NH4+ layers only improved the the NOx removal efficiency ~200-250 oC

Author Response

We thank the reviewer for his/her comments on our manuscript. Based on the reviewer’s comments, we have now made a few changes to the manuscript, which are listed below.
